# MOF-Derived Urchin-like Co$_9$S$_8$-Ni$_3$S$_2$ Composites on Ni Foam as Efficient Self-Supported Electrocatalysts for Oxygen Evolution Reaction

Yingping Bu [1,2], Yawen Zhang [2], Yingying Liu [2], Simin Li [2], Yanlin Zhou [2], Xuefen Lin [2], Zicong Dong [2], Renchun Zhang [2], Jingchao Zhang [2,*] and Daojun Zhang [2,*]

[1] College of Chemistry, Zhengzhou University, Zhengzhou 450001, China
[2] College of Chemistry and Chemical Engineering, Anyang Normal University, Anyang 455000, China
[*] Correspondence: zjc19830618@126.com (J.Z.); zhangdj0410@sohu.com (D.Z.); Tel.: +86-372-2900040 (D.Z.)

**Abstract:** Effective and inexpensive electrocatalysts are significant to improve the performance of oxygen evolution reaction. Facing the bottleneck of slow kinetics of oxygen evolution reaction, it is highly desirable to design the electrocatalyst with high activity, good conductivity, and satisfactory stability. In this work, nickel foam supported hierarchical Co$_9$S$_8$–Ni$_3$S$_2$ composite hollow microspheres were derived from in situ-generative MOF precursors and the subsequent sulfurization process by a simple two-step solvothermal method. The composite microspheres were directly grown on nickel foam without any binder, and nickel foam was used as the nickel source and support material. The morphology and constitution of the series self-supported electrodes were characterized by SEM, TEM, XRD, XPS, and Raman, respectively. The unique porous architecture enriched the electrode with sufficient active surface and helped to reactants and bubble evolved during electrochemical water oxidation. Through tuning the concentration of cobalt source and ligand, the content ratio of Co$_9$S$_8$ and Ni$_3$S$_2$ can be modulated. The heterostructures not only afford active interfaces between the phases but also allow electronic transfer between Co$_9$S$_8$ and Ni$_3$S$_2$. The optimized Co$_9$S$_8$-Ni$_3$S$_2$/NF-0.6 electrode with the highest electrochemical surface area and conductivity shows the best OER performance among the series electrodes in 1 M KOH solution. The overpotential of Co$_9$S$_8$-Ni$_3$S$_2$/NF-0.6 is only 233 mV when the current density is 10 mA cm$^{-2}$, and corresponding Tafel slope is 116.75 mV dec$^{-1}$. In addition, the current density of Co$_9$S$_8$-Ni$_3$S$_2$/NF-0.6 electrocatalyst hardly decreased during the 12 h stability measurement. Our approach in this work may provide the future rational design and synthesis of satisfactory OER electrocatalysts.

**Keywords:** oxygen evolution reaction; transition metal sulfides; MOFs; solvothermal; electrocatalysts



## 1. Introduction

With the growing environmental pollution crisis and rapid running out of traditional fossil fuels around the world, the researchers are devoted to develop clean alternative energies and explore efficient renewable energy conversion devices to meet the ever-increasing amount of requests [1–5]. In the last decade, various micro-/nanostructured transition-metal sulfides (TMSs) electrocatalysts have been prepared and demonstrated high-performances in water splitting, metal–air batteries, and fuel cells fields [6–13]. Oxygen evolution reaction (OER) electrocatalysts play a pivotal role in these oxygen-based renewable energy conversion technologies [14–16]. However, the OER has intrinsically rate-limiting, four electron–proton transfer steps and sluggish kinetic limitations. Therefore, additional energy is needed to overcome the high overpotential when electrocatalytically splitting water. In general, using high-performance electrocatalysts can accelerate the OER rates and reduce the voltage of water splitting in whole [17,18]. Noble metal oxides such as IrO$_2$ and RuO$_2$ still represent the state-of-the-art OER electrocatalysts up to date. However,

the rare reserves and expensive noble metal catalysts have hampered their large-scale application in oxygen-involved reaction energy conversion devices.

$Co_9S_8$ and $Ni_3S_2$ based TMSs have drawn extraordinary attention in electrochemical energy storage and conversion fields, such as electrocatalytic hydrogen production, super-capacitors, lithium-ion batteries, and rechargeable zinc–air batteries, due to their near-metal conductivity [19,20]. To date, various morphologies and structures of $Co_9S_8$ and $Ni_3S_2$, including nanoparticles, nanosheets, nanobelts, microplates, and composites, have been developed [21–26]. Although $Co_9S_8$ and $Ni_3S_2$ single-metal sulfides have been extensive researched, $Co_9S_8$-$Ni_3S_2$ composites are considered to be excellent electrocatalysts for OER ascribe to their low cost, non-toxic nature, and synergetic effect of multiple electrocatalytic sites [27–32]. The design and synthesis of $Co_9S_8$-$Ni_3S_2$ composites with robust structure, tunable composition and high conductivity will endow the electrode with outstanding properties [33,34].

Metal–organic frameworks (MOFs) are a novel class of porous organic-inorganic hybrid materials constructed by metal nodes and organic ligands, which demonstrate tunable porous structures and many potential applications [35,36]. In addition, MOF-derived transition metal micro-/nanomaterials, including transition metal oxides, chalcogenides, phosphides, and hydroxides, with adjustable structure and large BET surface area, have aroused considerable attention in energy storage and conversion fields [37–47]. In recent years, our group has been devoted to designing and synthesizing transition metal oxide and chalcogenide materials via MOF precursor conversion with different electrochemical performance such as nonenzymatic glucose electrochemical sensing, OER electrocatalysts, supercapacitors, and alkaline Ni–Zn batteries [48–50]. In this work, we demonstrate MOFs-derived $Co_9S_8$–$Ni_3S_2$ composites as electrode materials for efficient alkaline OER electrocatalysts. 1,3,5-benzenetricarboxylic acid was selected as rigid bridging ligand and earth-abundant transition metals as node to synthesize MOF precursors, and uniformly urchin-like MOF structures successfully grown on Ni foam substrate. Then, through secondary solvothermal sulfurization conversion, a series of $Co_9S_8$-$Ni_3S_2$ composites were successfully integrated on the Ni foam as self-supported electrodes and denoted as $Co_9S_8$-$Ni_3S_2$/NF-0.4, $Co_9S_8$-$Ni_3S_2$/NF-0.6 and $Co_9S_8$-$Ni_3S_2$/NF-0.8, respectively. Especially benefiting from the presence of abundant active surfaces and interfaces, short ion/electron transfer pathway, reinforced electronic transfer between $Co_9S_8$ and $Ni_3S_2$, the optimized $Co_9S_8$-$Ni_3S_2$/NF-0.6 hierarchical microspheres self-supported electrode displayed a much lower overpotential with 233 mV at 10 mA cm$^{-2}$ for OER, which is better than that of some recently reported TMS electrocatalysts.

## 2. Experimental

### 2.1. Reagents and Materials

Tetrabutylammonium bromide (TBAB), urea ($CO(NH_2)_2$), cobalt nitrate hexahydrate ($Co(NO_3)_2 \cdot 6H_2O$), ethylene glycol ($HOCH_2CH_2OH$), potassium hydroxide (KOH), and nickel foam (NF) were all purchased from Sinopharm Chemical Reagent CO., LTD. 1,3,5-benzenetricarboxylic acid ($C_9H_6O_6$, $H_3BTC$) was purchased from Shanghai Maclin Biochemical Technology Co., LTD. Ethanol ($C_2H_5OH$, 75%) and thioacetamide ($CH_3CSNH_2$, TAA) were purchased from Shanghai Aladdin Biochemical Technology Co., Ltd., Shanghai, China. Ethanol ($C_2H_5OH$, $\geq$ 99.7%) was purchased from Tianjin Fuyu Fine Chemical Co., Ltd., Tianjing, China. All chemicals are analytical grade and do not require further purification when used.

### 2.2. Synthesis of Electrocatalysts

Nickel foam was pretreated before use. Firstly, the NF was cut into $1 \times 3$ cm$^2$ size, and then in turn soaked in 3 M HCl and acetone for 30 min and 10 min, respectively. Then, ultrasonic cleaning was carried out for several times in deionized $H_2O$ and ethanol solution to remove impurities and oxides on the surface of NF.

Electrocatalyst is synthesized by a facile two-step solvothermal process. At first, 1.2 g urea, 0.1 g TBAB, 0.6 mM $Co(NO_3)_2 \cdot 6H_2O$ and $H_3BTC$, 10 mL $H_2O$ and 5 mL 75% ethanol solution were successively added to a 20 mL vial, and the mixture was then stirred vigorously for 30 min to form a uniform solution. After that, a piece of NF was immersed in the reaction solution and kept in an oven at 100 °C for 12 h. After the reaction, the NF was cleaned with deionized water and ethanol and dried in a vacuum drying oven at 60 °C to obtain the MOF precursor. For the sulfurization process, 0.2 mM TAA, 4 mL ethanol, 2 mL ethylene glycol and 1 mL $H_2O$ were successively added to a 20 mL vial and stirred for 30 min. After that, the MOF precursor covered NF was immersed in the mixture and heated at 160 °C for 8 h. After the reaction, the NF was washed with deionized water and ethanol and dried in a vacuum oven at 60 °C. Finally, $Co_9S_8$-$Ni_3S_2$/NF-0.6 electrocatalyst was obtained (Scheme 1). The preparation method of $Co_9S_8$-$Ni_3S_2$/NF-0.4 and $Co_9S_8$-$Ni_3S_2$/NF-0.8 is the same as that of $Co_9S_8$-$Ni_3S_2$/NF-0.6, only needs to adjust both $Co(NO_3)_2 \cdot 6H_2O$ and $H_3BTC$ to 0.4 mM and 0.8 mM, respectively. In addition, there is no change in the condition of second sulfurization stage, and the average mass increase of $Co_9S_8$-$Ni_3S_2$/NF series electrodes are 4~5 mg. In order to investigate the influence of Co ions in the morphology of precursors, as a comparison, $Ni_3S_2$/NF electrode was prepared without $Co(NO_3)_2 \cdot 6H_2O$ in the first solvothermal step and kept other conditions consistent with the synthesis procedure of $Co_9S_8$-$Ni_3S_2$/NF-0.6, and the mean loading weight is 3~4 mg.

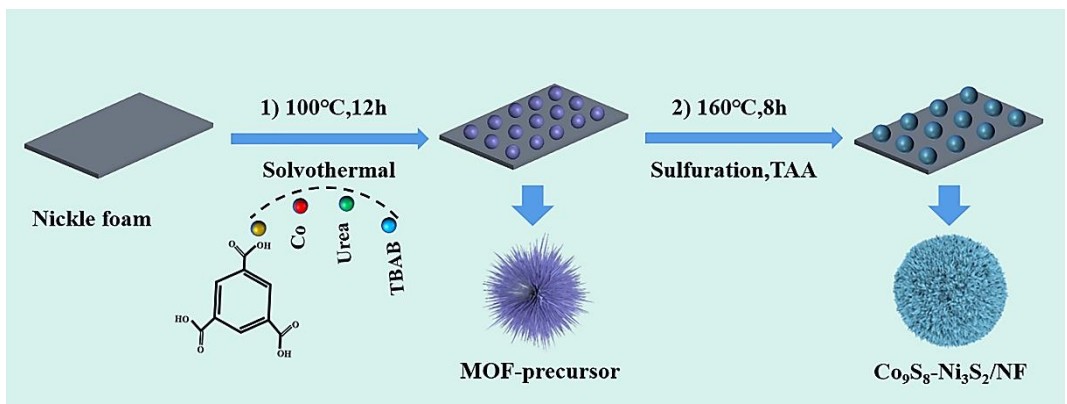

**Scheme 1.** The illustration of the synthesized process of $Co_9S_8$-$Ni_3S_2$/NF electrodes.

### 2.3. Materials Characterization

The phases of the electrocatalyst were characterized by Ultima III X-ray powder diffractometer (XRD) and Thermo Scientific K-Alpha X-ray photoelectron spectroscopy (XPS). Field emission scanning electron microscope (FE-SEM), Tecnai $G^2S$-Twin F20 field emission transmission electron microscope (TEM) and energy dispersive X-ray spectroscopy (EDS) analyzer were used to characterize the morphologies and composition of the electrodes. Raman spectroscopy was conducted by LabRAM HR Evol Raman microscopy under 532 nm laser excitation.

### 2.4. Electrochemical Measurements

Electrochemical tests were performed in 1 M KOH solution and conducted on a CHI 760E electrochemical workstation using a standard three-electrode system at room temperature. $Co_9S_8$-$Ni_3S_2$/NF-0.4, -0.6, -0.8, and $Ni_3S_2$/NF were used as working electrodes, and their working geometric area under the electrolyte solution level is 1 cm$^2$. Hg/HgO electrode was used as the reference electrode, and platinum mesh (1 cm$^2$) was the counter electrode. The electrolyte was treated with $O_2$ for 30 min before the test to ensure that the electrolyte solution is oxygen saturated. The linear sweep voltammetry curve was obtained at a sweep speed of 2 mV·s$^{-1}$. The Tafel slope is obtained from the Tafel equation ($\eta$ = b log$j$ + a, where a is a constant, $j$ is the current density, and b is the Tafel slope). The

electrochemical impedance was measured from $10^5$ to 0.1Hz with an amplitude potential of 5 mV. The electrochemical active area (ECSA) was proportional to the double layer capacitance ($C_{dl}$), which was evaluated by CV curves with various scan rates (5 to 30 mV s$^{-1}$) in the potential range from 0.1 to 0.2 V vs. Hg/HgO. Multicurrent measurements and long-term stability measurements were also performed in 1 M KOH solution to assess the stability of the electrocatalyst. The potential in this experiment was converted into reversible hydrogen potential, and the calculation formula is $E_{RHE} = E_{Hg/HgO} + (0.059 \times pH + 0.098 V)$. It is important to note that none of the potentials in this experiment were compensated by IR.

## 3. Results and Discussion

PXRD diffraction patterns were used to analyze the crystal phase of the synthesized series of electrodes. Figure 1 represents the PXRD patterns and their standard JCPDS profiles for $Ni_3S_2$/NF (NS), $Co_9S_8$-$Ni_3S_2$/NF-0.4 (CNS-0.4), $Co_9S_8$-$Ni_3S_2$/NF-0.6 (CNS-0.6), and $Co_9S_8$-$Ni_3S_2$/NF-0.8 (CNS-0.8) electrodes, respectively. It can be observed from Figure 1 that the synthesized four electrodes all have two intense and sharp diffraction peaks at 2θ of 44.4° and 51.9°, which correspond to the Ni foam, while the three electrodes with different Ni/Co ratios showed similar diffraction peaks, all of which can correspond well with $Co_9S_8$ (JCPDS No. 00-019-0364) and $Ni_3S_2$ (JCPDS No. 01-073-0698) phases. The characteristic diffraction peaks of CNS-0.4, CNS-0.6, and CNS-0.8 at 2θ of 15.4°, 29.8°, 39.4°, 47.3° and 61.3° are attributed to the lattice planes (111), (311), (331), (511) and (533) of $Co_9S_8$, respectively. On the other side, the characteristic diffraction peaks at 21.8°, 30.9°, 31.1°, 37.8°, 49.8° and 55.3° belong to the crystal planes (100), (110), (1–10), (111), (210) and (333) of $Ni_3S_2$, respectively. The results confirmed the formation of $Co_9S_8$ and $Ni_3S_2$ phases on the NF of these catalysts. The diffraction peak intensity of $Co_9S_8$ phase can be optimized by the amount of cobalt ions added.

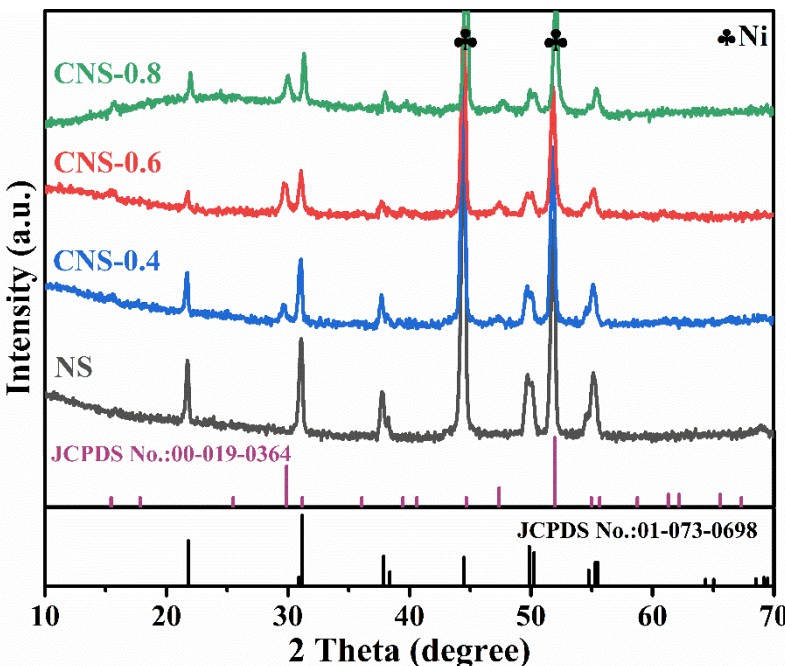

**Figure 1.** PXRD patterns of $Ni_3S_2$/NF, $Co_9S_8$–$Ni_3S_2$/NF-0.4, $Co_9S_8$–$Ni_3S_2$/NF-0.6, and $Co_9S_8$–$Ni_3S_2$/NF-0.8 electrodes.

X-ray photoelectron spectroscopy (XPS) was used to analyze the surface chemical composition and elemental valence of CNS-0.4, -0.6, -0.8 samples. The measured scanning spectra in Figure 2a shows that there are three elements Co, Ni and S in CNS-0.4, -0.6, and -0.8 samples. The Co 2p high-resolution spectrum can be divided into two spin-orbit doublet peaks and two satellite peaks, as shown in Figure 2b. The binding energies at 780.0



and 795.1 eV correspond to $2p_{3/2}$ and $2p_{1/2}$ of $Co^{3+}$, respectively. The binding energies at 781.4 and 797.8 eV correspond to $2p_{3/2}$ and $2p_{1/2}$ of $Co^{2+}$, respectively. The binding energies at 787.0 and 803.5eV are satellite peaks [29,30] Compared with CNS-0.4 and CNS-0.8 samples, the dominant valence state of Co ions in CNS-0.6 sample is +2. Figure 2c displays the high-resolution XPS spectra in the Ni 2p region, and the three sets of peaks obtained correspond to two spin-orbit peaks and two satellite peaks, respectively. The peaks at 852.0 and 872.8 eV correspond to $2p_{3/2}$ and $2p_{1/2}$ of $Ni^{2+}$, respectively. The peaks at 855.8 and 874.3 eV correspond to $2p_{3/2}$ and $2p_{1/2}$ of $Ni^{3+}$, respectively. The peaks at 861.6 and 879.6 eV are satellite peaks [31,32]. In addition, the Ni 2p XPS result also indicated the $Ni^{3+}/Ni^{2+}$ peak area ratios of the CNS-0.6 sample are largest among the three samples, which indicated that the proper Co doping may modulate the electronic structure of $Co_9S_8$-$Ni_3S_2$ composites. The S 2p spectrum in Figure 2d shows that there are three peaks at the binding energies of 161.1, 162.2 and 163.3 eV, corresponding to the M-S bond in $S_2^{2-}$ (M is Ni, Co), which is the sulfide ion in the electron transport between $Ni_3S_2$ and $Co_9S_8$ nanostructures, and the peak at 169.8 eV is the satellite peak [33].

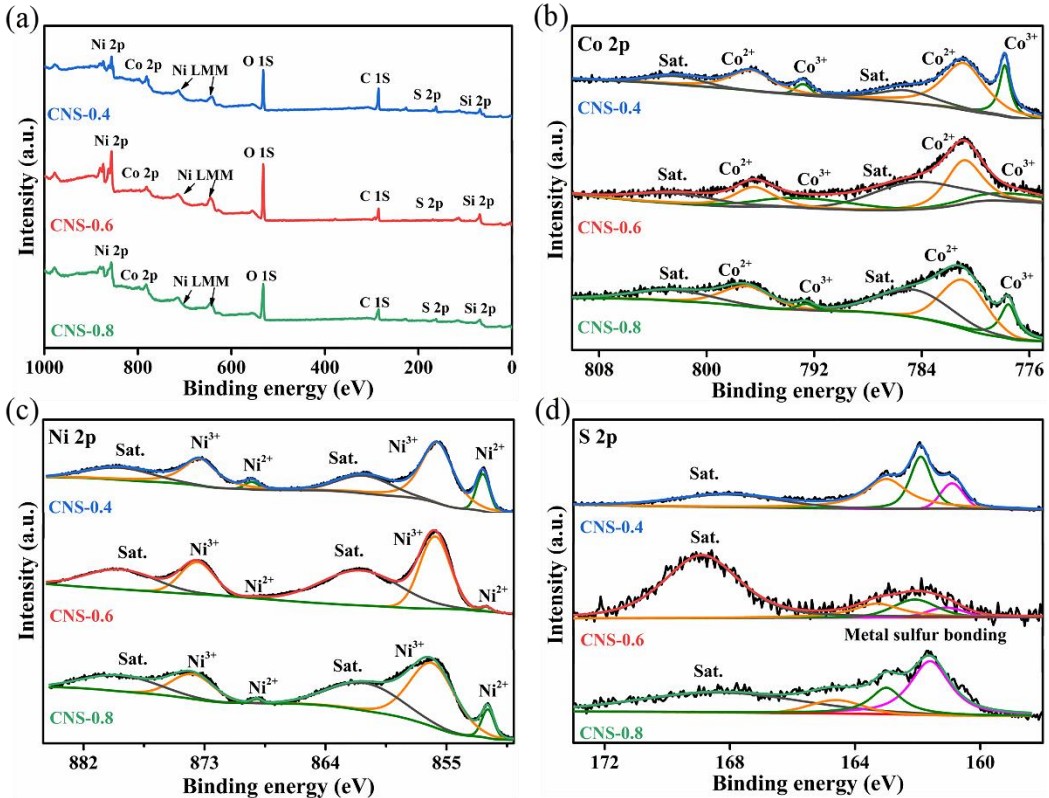

**Figure 2.** XPS of $Co_9S_8$–$Ni_3S_2$/NF-(0.4, 0.6, 0.8), (**a**) survey spectra, high-resolution spectra of (**b**) Co 2p, (**c**) Ni 2p, and (**d**) S 2p.

Raman spectra were also utilized to examine the composition of the NS, CNS-0.4, CNS-0.6, and CNS-0.8 samples grown on Ni foam directly (Figure S1). The peaks located in 202, 223, 304, 321, and 350 cm$^{-1}$, which corresponded to the characteristic peaks of $Ni_3S_2$ phase [51,52]. Moreover, the samples of CNS-0.4, CNS-0.6, and CNS-0.8 have new peaks centered at 373, 450, 517, and 673 cm$^{-1}$, indicating the existence of $Co_9S_8$ phase [53,54].

SEM technique is used to study the morphology of as-synthesized materials, Figures S2 and 3 shows the images of the MOF precursors of (a, d) CNS-0.4, (b, e) CNS-0.6, and (c, f) CNS-0.8 after the first step of solvothermal synthesis. As can be observed from the Figure S2, nickel foam covered with microsphere precursors. The further magnified SEM images in Figure 3 shows that the microspheres were composed of many willow-leaf-like nanosheets, and the average diameter of the microspheres was around 2.33–4.39 μm

(Figure S3). For comparison, the images of Ni$_3$S$_2$/NF sample without Co ions formed rough nanosheets on the surface of nickel foam (Figure S4).

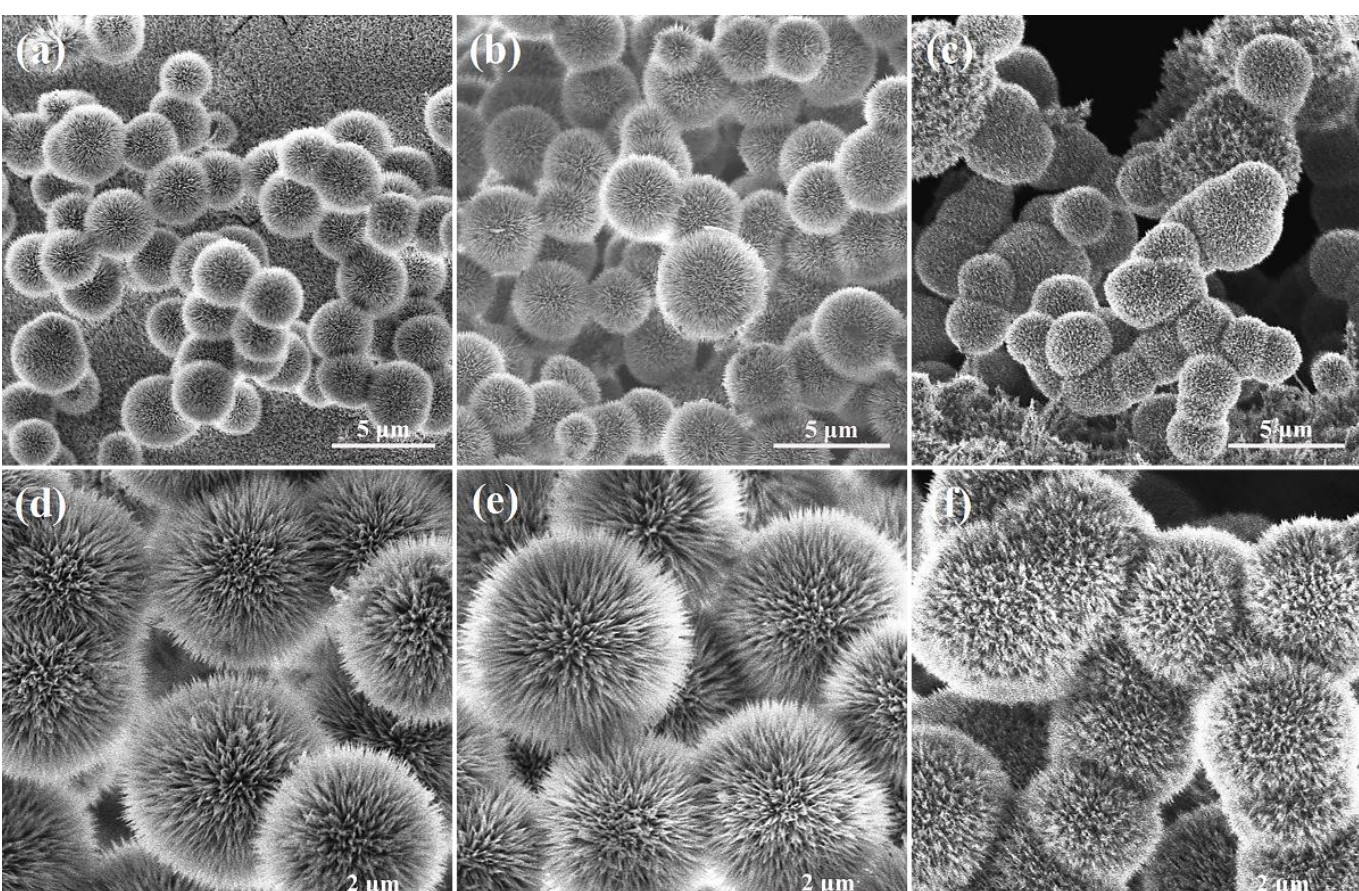

**Figure 3.** SEM images of MOF precursors for (**a**,**d**) Co$_9$S$_8$-Ni$_3$S$_2$/NF-0.4, (**b**,**e**) Co$_9$S$_8$-Ni$_3$S$_2$/NF-0.6, and (**c**,**f**) Co$_9$S$_8$-Ni$_3$S$_2$/NF-0.8.

SEM images and EDS elemental mapping of CNS-0.4, -0.6, -0.8 are exhibited in Figure 4. Compared with the MOF precursors, the morphology of the MOF-derived samples, CNS-0.4 (Figure 4a–c), CNS-0.6 (Figure 4d–f), and CNS-0.8 (Figure 4g–i), still maintained the morphology of microspheres, but the average diameter increased significantly due to agglomeration, ranging from 3.11–9.08 μm (Figure S3). There are many small nanoparticles on the surface of the nanosheets, which may be the building block of nanosheets, that make up the microspheres, whose surface is very rough. At the same time, it can also be observed from Figure 4e that these microspheres are hollow inside. The unique hollow spherical structure and rough surface are not only conducive to the exposure of the catalytic active sites, but also can further accelerate the electrolyte penetration and OER reaction kinetics.

Furthermore, SEM-EDX element mapping test was also carried out on Co$_9$S$_8$-Ni$_3$S$_2$/NF series samples, and the results were shown in Figure 4c,f,i, respectively. All the as-synthesized materials are composed of Co, Ni and S elements, and these elements are uniformly distributed. Figure S5 shows the EDS patterns of the samples, in which the Co content of Co$_9$S$_8$-Ni$_3$S$_2$/NF-0.6 is higher than the other two samples, which is consistent with the PXRD test results.

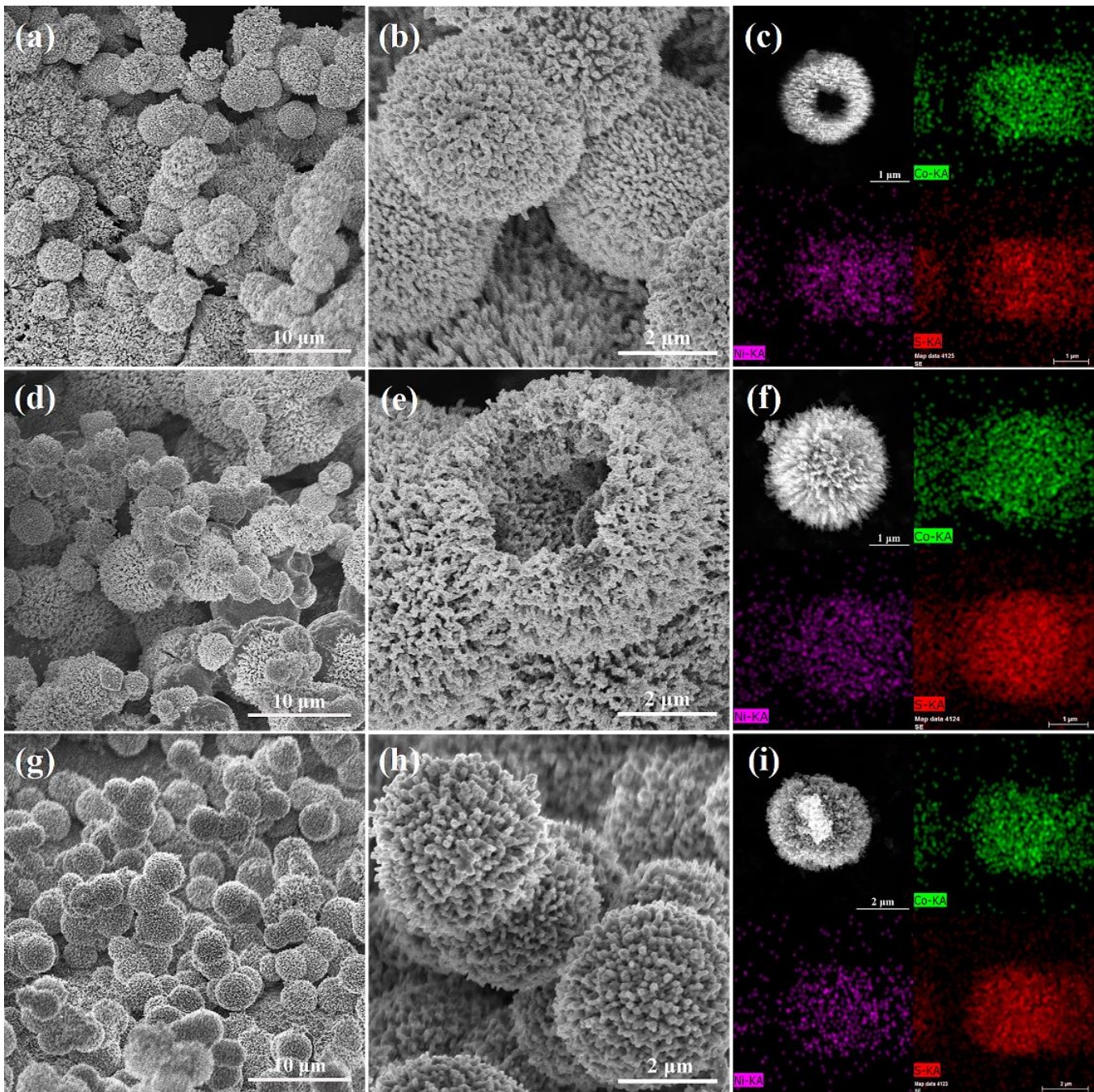

**Figure 4.** SEM images and EDS elemental mapping of series electrodes, (**a**–**c**) $Co_9S_8$-$Ni_3S_2$/NF-0.4 with different magnification. (**d**–**f**) $Co_9S_8$–$Ni_3S_2$/NF-0.6, (**g**–**i**) $Co_9S_8$–$Ni_3S_2$/NF-0.8.

Figure 5a shows the low-magnification TEM image of CNS-0.6 via ultrasonic stripping from Ni foam. HRTEM image in Figure 5b show clear lattice fringes, with the crystal plane spacing of 0.552 nm and 0.298 nm attributed to (111) and (311) planes of $Co_9S_8$, and the crystal plane spacing of 0.289 nm attributed to (110) plane of $Ni_3S_2$. The results are in accordance with the PXRD patterns exhibited in Figure 1. Figure 5c is the HAADF-STEM element mapping images of CNS-0.6. It can also be seen from the image that the morphology of the sample is a microsphere composed of many thin lance-shaped nanosheets, and the three elements of Co, Ni and S are evenly distributed in the sample. From the HRTEM result, it can be inferred that the upper layer is $Co_9S_8$ and the bottom is $Ni_3S_2$. This result is mainly supported by Figure 5b.

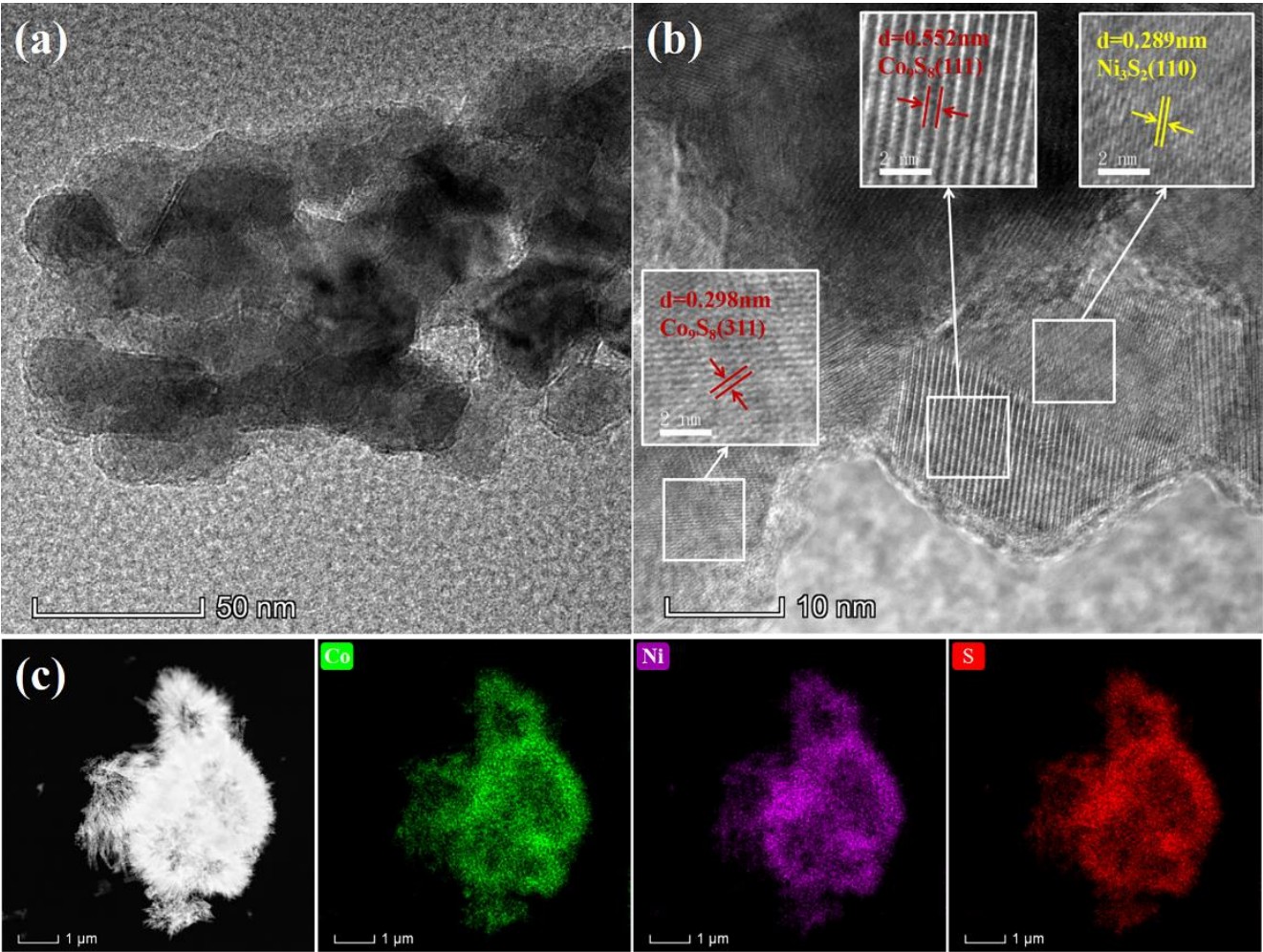

**Figure 5.** The low (**a**) and high-resolution (**b**) TEM images of $Co_9S_8$–$Ni_3S_2$/NF-0.6 sample. (**c**) HAADF-STEM images of the $Co_9S_8$–$Ni_3S_2$/NF-0.6 and the corresponding EDX mapping of Co, Ni, and S elements.

The effect of electrocatalysts with different cobalt content on the OER performance was investigated in $O_2$-saturated 1.0 M KOH electrolyte solution and the relevant results are displayed in Figure 6. Figure 6a shows the LSV curves of all the electrodes towards OER. The $Ni_3S_2$/NF shows weak catalytic activity toward OER, while CNS-0.6 shows clearly the highest activity compared to the other electrocatalysts. The CNS-0.6 only needs a very small overpotential ($\eta$ = 233 mV) to drive a current density of 10 mA cm$^{-2}$. This value is outperformed than that of NS ($\eta_{10}$ = 381 mV), CNS-0.4 ($\eta_{10}$ = 275 mV) and CNS-0.8 ($\eta_{10}$ = 247 mV) electrodes, and even lower than previously reported catalysts such as Fe–$Co_9S_8$ NM/NF [55], Ni–Co–S/NSC [56], 3D $Ni_3S_2$/NF-4 [57], NiS/NiFe2O4 [58], Ni–$Co_3S_4$-2 nanospheres [59], $Ni_3S_2$–$Co_9S_8$/NF [60], NiCo2S4 [61], and $Co_9S_8$ NTs/Ni [62] (Table 1). Evidently, the Co incorporation can significantly increase the activity of OER performance compared with $Ni_3S_2$/NF sample with the absence of Co element. In addition, the effective regulation of Co ions doping amount have great influence on the electrocatalytic activity of synthesized $Co_9S_8$–$Ni_3S_2$/NF series samples. Moreover, compared with the other three electrodes, CNS-0.6 has the largest current density in the measured potential range, which proves that it has excellent catalytic performance for OER.

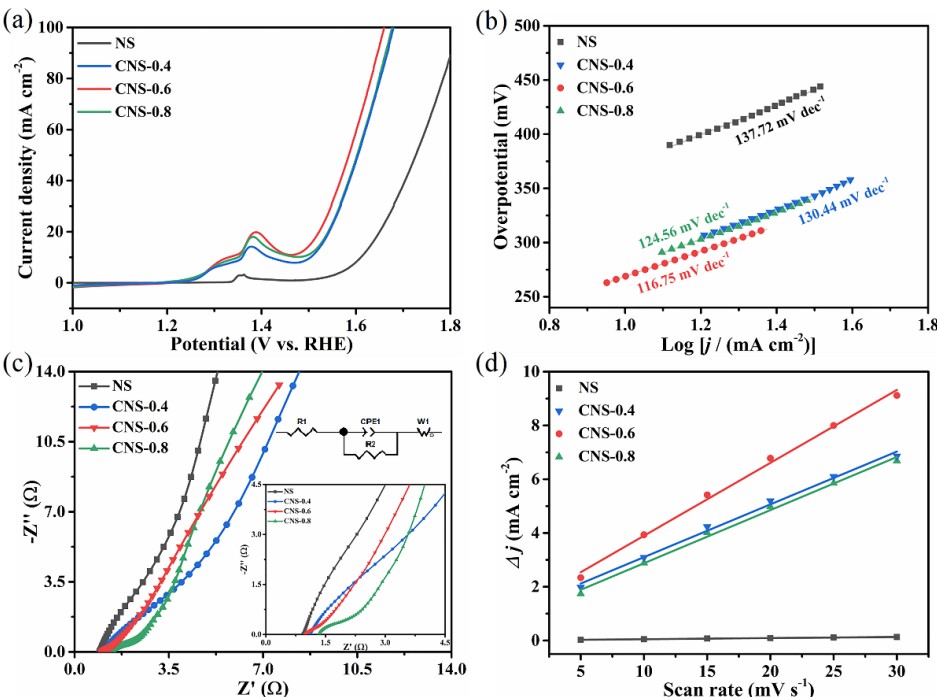

**Figure 6.** (**a**) The LSV curves of series electrodes at a scan rate of 2 mV s$^{-1}$ in 1 M KOH solution; (**b**) The corresponding Tafel plots; (**c**) Nyquist plots; (**d**) C$_{dl}$ at different scan rates. Capacitive currents at 0.15 V (vs. Hg/HgO) ($\Delta j = j_a - j_c$).

**Table 1.** Comparison of the catalytic properties of Co$_9$S$_8$-Ni$_3$S$_2$/NF-0.6 with reported catalysts towards OER performances in alkaline medium.

| Catalysts | Overpotential (mV) | Tafel Slope (mV dec$^{-1}$) | Ref. |
|:---:|:---:|:---:|:---:|
| Fe-Co$_9$S$_8$ NM/NF | $\eta_{10} = 270$ | 70.0 | [55] |
| Ni–Co-S/NSC | $\eta_{10} = 309$ | 87.0 | [56] |
| 3D Ni$_3$S$_2$/NF-4 | $\eta_{10} = 242$ | 76.0 | [57] |
| NiS/NiFe$_2$O$_4$ | $\eta_{10} = 230$ | 88.0 | [58] |
| Ni–Co$_3$S$_4$-2 nanospheres | $\eta_{10} = 298$ | 90.5 | [59] |
| Ni$_3$S$_2$–Co$_9$S$_8$/NF | $\eta_{20} = 294$ | 80.0 | [60] |
| NiCo$_2$S$_4$ | $\eta_{10} = 260$ | 55.0 | [61] |
| Co$_9$S$_8$ NTs/Ni | $\eta_{50} = 394$ | 136.4 | [62] |
| | $\eta_{10} = 224$ | | |
| Co$_9$S$_8$–Ni$_3$S$_2$/NF-0.6 | $\eta_{10} = 233$ | 116.75 | This work |
| | $\eta_{20} = 292$ | | |
| | $\eta_{50} = 355$ | | |

The reaction kinetics for OER was further discussed by Tafel slope. Figure 6b displays the Tafel slope of NS, CNS-0.4, CNS-0.6 and CNS-0.8, which were calculated as 137.72, 130.44, 116.75 and 124.56 mV dec$^{-1}$, respectively. The reason for the large Tafel slope values is due to the strong redox peak in testing the LSV curves. It turns out that the Tafel of CNS-0.6 is relatively small compared to the other samples, indicating its outstanding OER dynamics. Electrochemical impedance spectroscopy (EIS) was also used to evaluate the charge transfer resistance of all catalysts, and Nyquist plots of all electrodes were shown in Figure 6c. It can be concluded that the semicircle diameter of CNS-0.6 is smaller than that of the other electrodes, and its measured $R_{ct}$ value 0.47 Ω is the smallest among of the four electrodes. In contrast, NS exhibits the largest $R_{ct}$ value of 58.10 Ω, the $R_{ct}$ values of CNS-0.4 and CNS-0.8 are 37.31 Ω and 1.03 Ω, respectively. It is proved that the charge transfer ability of CNS-0.6 on the electrode surface and electrolyte is relatively excellent. This result may be related to the Co doping content. With appropriate Co doping content, the electron transfer rate and resistance value are optimized, and the OER activity is satisfactory.

In order to evaluate the true catalytic activity of the as-synthesized electrodes, it is necessary to analyze the surface area that is actually involved in the electrochemical catalytic reaction. The ECSA is proportional to the capacitance of the double electric layer ($C_{dl}$). $C_{dl}$ values were obtained by performing CV measurements in the non-Faraday region, and the CV curves measured at different scan rates (5–30 mV s$^{-1}$) were shown in Figure S6. The liner slopes were obtained by linear fitting of the current density at 0.15 V and scan rate, which correspond to the $2C_{dl}$ values of all the samples. The measured $C_{dl}$ values of CNS-0.6, CNS-0.8, CNS-0.4 and NS are 135.59, 98.82, 98.20 and 2.08 mF cm$^{-2}$, respectively (Figure 6d). This result indicates that CNS-0.6 exhibits the largest electrochemical active surface area among all the four electrodes. This indicates the CNS-0.6 catalyst exposed more active sites and showed excellent OER activity when appropriate Co content doped.

The practicability of electrocatalysts needs to be investigated by measuring the stability of the materials. Figure 7a shows multistep chronopotentiometric curves for all synthesized materials tested at current densities ranging from 10 to 100 mA cm$^{-2}$. It can be seen that, in each step of the test, the current density increases by 10 mA cm$^{-2}$ every 500 s while the potential tends to be stable. At the same time, with the gradual increase of the current density, the order height of the potential also gradually decreases. In addition, the working potential of CNS-0.6 is the lowest in all electrodes, indicating it maintained well stability in a wide range of current density, and the mass diffusion rate is relatively high. Figure 7b exhibits the long-term stability test of these catalysts at a potential of 0.7 V (vs. Hg/HgO). The current density of CNS-0.6 remained at 50 mA cm$^{-2}$ during the electrolysis process up to 12 h, and the current density almost did not decay. Figure 7c shows the polarization curves of CNS-0.6 electrode before and after accelerated cycle stability test for 1000 cycles, the two polarization curves basically coincided, which further verified the superior OER stability of CNS-0.6 electrode. SEM images of the electrodes after long-term OER stability test were shown in Figure S7, it can be observed from the figure that all the $Co_9S_8$-$Ni_3S_2$/NF samples maintain the morphology of the microspheres. The XPS after stability test were displayed in Figure S8, take the CNS-0.6 electrode as an example, the $Co^{3+}$ peaks at 778.1 and 792.5 eV were enhanced after stability test, which shown an opposite trend of CNS-0.4 and CNS-0.8 electrodes, indicating that the valence state of cobalt ion has changed in the long-term stability test, and increased $Co^{3+}$ is good for charge transfer process and is contributed to the high activity of the CNS-0.6 electrode [63,64]. In addition, the diffraction peaks in the PXRD patterns did not change after the OER stability test (Figure S9), except the intensity of the diffraction peaks of $Co_9S_8$ fade, also indicating that the incorporation of Co element played a key role in catalytic activity.

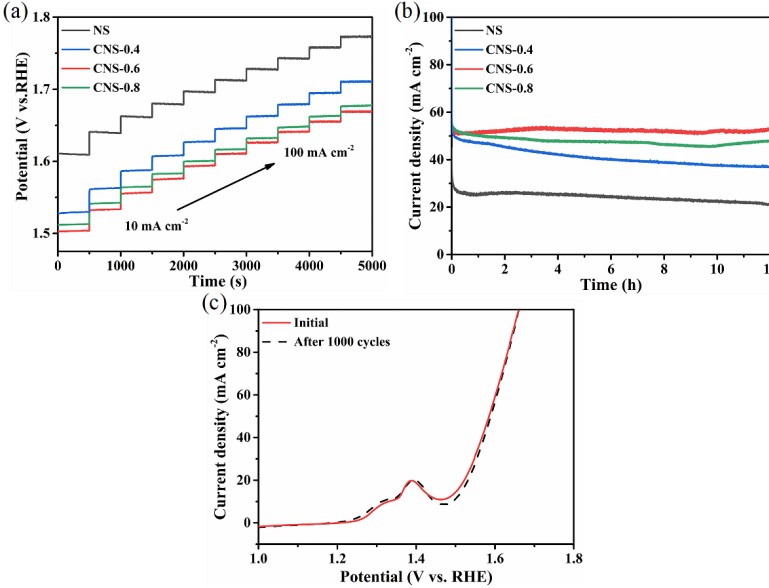

**Figure 7. (a)** Multicurrent process curves for $Ni_3S_2$/NF, $Co_9S_8$-$Ni_3S_2$/NF-0.4, $Co_9S_8$-$Ni_3S_2$/NF-0.6 and

$Co_9S_8$-$Ni_3S_2$/NF-0.8 electrodes with the current density range of 10–100 mA cm$^{-2}$. (**b**) OER stability measurements for $Ni_3S_2$/NF, $Co_9S_8$-$Ni_3S_2$/NF-0.4, $Co_9S_8$-$Ni_3S_2$/NF-0.6 and $Co_9S_8$-$Ni_3S_2$/NF-0.8 at a potential of 0.7 V (vs. Hg/HgO) for 12 h. (**c**) The polarization curves of $Co_9S_8$-$Ni_3S_2$/NF-0.6 electrode before and after accelerated cycle stability test for 1000 cycles.

## 4. Conclusions

An excellent self-supported electrocatalyst for OER was developed by in situ growth of $Co_9S_8$-$Ni_3S_2$ composite urchin-like hollow microspheres on nickel foam via a facile solvothermal sulfurization conversion method. Meanwhile, the MOF precursor derivative synthetic method is simple and low power consuming. The constructed self-supported electrode without any binder has unique pore and hierarchical morphology, which is conducive to electrolysis process. The optimized self-supported electrode, denoted as $Co_9S_8$-$Ni_3S_2$/NF-0.6, showed the highest OER activity in alkaline solution. The OER overpotential of the self-supported electrode at 10 mA cm$^{-2}$ is 233 mV. There is no decay of current density for $Co_9S_8$-$Ni_3S_2$/NF-0.6 electrode during the stability evaluation for 12 h. The as-prepared urchin-like $Co_9S_8$-$Ni_3S_2$ composite self-supported electrode with modulated electronic structure and the presence of heterointerfaces provides intrinsic OER activity and facilitates the electron/mass transport. The porous hollow features can provide large electrochemical surface area and further accelerate the OER reaction kinetics. The structural integrity of the electrode also endows it with improved excellent catalytic stability. Our research may provide the rational design and synthesis of satisfactory OER electrocatalysts in future.

**Supplementary Materials:** The following supporting information can be downloaded at: https://www.mdpi.com/article/10.3390/batteries9010046/s1, Figure S1: Raman spectra of series electrodes; Figure S2: SEM images of series electrodes; Figure S3: Particle size distribution images of series samples; Figure S4: SEM images of precursor of $Ni_3S_2$/NF, after vulcanization and stability test; Figure S5: EDS diagrams of $Co_9S_8$-$Ni_3S_2$ composites; Figure S6: CV plots of $Ni_3S_2$/NF, $Co_9S_8$-$Ni_3S_2$/NF-0.4, -0.6 and -0.8 electrodes with different scan rates; Figure S7: SEM images of series electrodes after stability test; Figure S8: XPS images of $Co_9S_8$-$Ni_3S_2$/NF-0.4, -0.6 and -0.8 after OER stability test; Figure S9: PXRD patterns after OER stability test.

**Author Contributions:** Author Conceptualization, D.Z.; Data curation, Y.B., Y.Z. (Yawen Zhang), X.L. and J.Z.; Formal Analysis, Y.B., D.Z.; Funding acquisition, J.Z. and D.Z.; Investigation, Y.B., Y.Z. (Yanlin Zhou), Y.L., S.L., Z.D. and J.Z.; Methodology, R.Z. and J.Z.; Project administration, J.Z. and D.Z.; Supervision, J.Z. and D.Z.; Writing—original draft, Y.B., D.Z. and D.Z.; Writing—review & editing, Y.B., D.Z. and J.Z. All authors have read and agreed to the published version of the manuscript.

**Funding:** This work was supported the Science and Technology Research Project of Henan Province (222102240096), the Foundation of Henan Educational Committee (22A150002), the Program for Innovative Research Team of Science and Technology in the University of Henan Province (18IRT-STHN006), and by the National Science Foundation of China (No. 21603004).

**Institutional Review Board Statement:** Not applicable.

**Informed Consent Statement:** Not applicable.

**Data Availability Statement:** Not applicable.

**Conflicts of Interest:** The authors declare no conflict of interest.

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
