# Peer review of "MOF-Derived Urchin-like Co9S8-Ni3S2 Composites on Ni Foam as Efficient Self-Supported Electrocatalysts for Oxygen Evolution Reaction"

_batteries, doi:10.3390/batteries9010046_

Round 1

Reviewer 1 Report

The authors present work on MOF-Derived Urchin-like Co9S8-Ni3S2 Composites on Ni Foam as Efficient self-supported Electrocatalysts for Oxygen Evolution Reaction. The authors have carried out systematic investigation. The results are interesting and well-supported by experimental/analytical evidence. Therefore, I would recommend this article for publication in the Batteries after some major revisions as advised in my comments below:

-I feel that Abstract is needed with key point of characterization result. Please add some significant highlight from materials properties.

-Please enrich the introduction with the help of these papers’ citation https://doi.org/10.1016/j.inoche.2022.110136, https://doi.org/10.1016/j.matchemphys.2022.126861.

- The aim of the study is so clear, please discuss/add details.

-Why the author has selected two sulfides in one system?

-Why do authors only vary the Co content? Why not for Ni content?

-There are so many works related to the present work like https://doi.org/10.1016/j.jssc.2018.12.004 and others.

-PXRD: Why the peak intensity of the main peak is decreasing and emerging of additional peaks with increasing of m in details.

-XPS: The samples contained a substantial amounts of C,O. Please comment on it.

-“In addition, the effective regulation of Co ions doping amount has a great influence on the electrocatalytic activity of synthesized Co9S8-Ni3S2/NF series samples..” Authors need justification.

-SEM: how the authors recognised the indications? The quality of the images is not so good.

- The authors need to justify the compositions with analysis like XPS.

-There are some typos and grammatical errors in the text. Please revise this

In the abstract “electrocatalyst hardly decrease during” it would be decreased.

Author Response

Reviewer 1

Comments and Suggestions for Authors

The authors present work on MOF-Derived Urchin-like Co9S8-Ni3S2 Composites on Ni Foam as Efficient self-supported Electrocatalysts for Oxygen Evolution Reaction. The authors have carried out systematic investigation. The results are interesting and well-supported by experimental/analytical evidence. Therefore, I would recommend this article for publication in the Batteries after some major revisions as advised in my comments below:

-I feel that Abstract is needed with key point of characterization result. Please add some significant highlight from materials properties.

Response: The key point of characterization result has been added in Abstract in the revised version.

-Please enrich the introduction with the help of these papers’ citation https://doi.org/10.1016/j.inoche.2022.110136, https://doi.org/10.1016/j.matchemphys.2022.126861.

Response: The above articles have been cited in the revised version (Ref. 5 and 47).

- The aim of the study is so clear, please discuss/add details.

Response: We added some expression in the revised manuscript.

-Why the author has selected two sulfides in one system?

Response: Ni3S2 and Co9S8 are expected candidate for OER with high conductivity. The composite of Co9S8-Ni3S2 not only displayed synergistic effect but also showed higher electrochemical activity. The as-prepared urchin-like Co9S8-Ni3S2 composite self-supported electrode with modulated electronic structure and the presence of multicomponent interfaces provides intrinsic OER activity and facilitates the electron/mass transport. The composite electrodes are not only conducive to the exposure of the catalytic active sites, but also can further accelerate the electrolyte penetration and OER reaction kinetics. The structural integrity of the electrode also endows it with improved excellent catalytic stability.

-Why do authors only vary the Co content? Why not for Ni content?

Response: In this work, nickel foam not only serves as the support material, but also provides nickel raw material. We did not add additional nickel raw materials in the reaction system. The Co content can be tuned by the ratio of reactants, the Ni content is from Ni foam, thus, the reacting amount of Ni from substrate cannot be tuned.

-There are so many works related to the present work like https://doi.org/ 10.1016/j.jssc.2018.12.004 and others.

Response: The above article (Ref. 34) and some related works have been cited in the revised version.

-PXRD: Why the peak intensity of the main peak is decreasing and emerging of additional peaks with increasing of m in details.

Response: The peaks intensity of Ni3S2 decreased along with the increased amount of Co source and ligand in the first solvothermal step. The more MOF precursors generated in the first step, the less of uncover surface of Ni foam left. Thus, from the XRD pattern, there is a downward trend of the main peaks of Ni3S2 with the increasing amount of Co9S8.

-XPS: The samples contained a substantial amounts of C, O. Please comment on it.

Response: The C may be from the reference, and the O may be ascribed to the surface partial oxidation of Co9S8-Ni3S2/NF when exposed in air.

-“In addition, the effective regulation of Co ions doping amount has a great influence on the electrocatalytic activity of synthesized Co9S8-Ni3S2/NF series samples..” Authors need justification.

Response: The Co ions doping amount has a great influence in electrocatalytic activity mainly in two aspects. Firstly, the conductivity of the series samples was affected by Co ions doping amount, among the series samples, Co9S8-Ni3S2/NF-0.6 sample display the smallest charge transfer resistance. Secondly, Co9S8-Ni3S2/NF-0.6 sample shows the most uniform morphology, which endowed the sample with the largest electrochemical surface areas. As a consequence, Co9S8-Ni3S2/NF-0.6 shows extremely the highest activity compared to the other electrocatalysts.

-SEM: how the authors recognised the indications? The quality of the images is not so good.

Response: The quality of the SEM images has been improved in the revised version.

- The authors need to justify the compositions with analysis like XPS.

Response: We tried to justify the compositions in the revised manuscript.

-There are some typos and grammatical errors in the text. Please revise this.

Response: The typos and grammatical errors in the text has been corrected in the revised manuscript.

In the abstract “electrocatalyst hardly decrease during” it would be decreased.

Response: “electrocatalyst hardly decrease during” was changed to “decreased”.

Reviewer 2 Report

     In this manuscript, “MOF-Derived Urchin-like Co9S8-Ni3S2 Composites on Ni Foam 1 as Efficient self-supported Electrocatalysts for Oxygen Evolution Reaction”, the authors demonstrate the preparation of microspheres consisting of Co9S8-Ni3S2 composite materials and investigate their electrochemical performance as electrocatalysts for OER. Though the manuscript is suitable for publication in the special issue of the journal “Batteries”, the following revisions should be carried out to further improve the quality.

1.      In Figure 2, the lengthy figure captions such as ‘Co9S8-Ni3S2/NF-0.1’ and so on, make the figures less appealing. A short abbreviation should be replaced in the revised manuscript for better data presentation.

2.      The prepared composite materials should further be analyzed by Raman spectroscopy to confirm the successful conversion of starting materials and the different phases of individual components.

3.      The performance of electrocatalysts depends on the active mass. The values should be mentioned in the experimental section of the revised manuscript.

4.      The upper limits of X- & Y-axes of the EIS plot (Figure 6c) should be identical, for example, 8 (Ohm) in both axes. Further, the zoomed portion of EIS also should be included as an inset.

5.      The figure quality should be improved in the revised manuscript, especially, Figure 1, 2, 6 and 7.

6.      Plenty of cobalt & nickel sulfide-based electrocatalysts are reported in the literature. What are the advantages of the synthesized electrocatalysts, in terms of properties and electrocatalytic activity? These points should clearly be mentioned in the revised manuscript.

7.      The conclusion part of the manuscript is short. The important findings should be included in the revised manuscript.

Author Response

Reviewer 2

Comments and Suggestions for Authors

     In this manuscript, “MOF-Derived Urchin-like Co9S8-Ni3S2 Composites on Ni Foam 1 as Efficient self-supported Electrocatalysts for Oxygen Evolution Reaction”, the authors demonstrate the preparation of microspheres consisting of Co9S8-Ni3S2 composite materials and investigate their electrochemical performance as electrocatalysts for OER. Though the manuscript is suitable for publication in the special issue of the journal “Batteries”, the following revisions should be carried out to further improve the quality. 

  1. In Figure 2, the lengthy figure captions such as ‘Co9S8-Ni3S2/NF-0.1’ and so on, make the figures less appealing. A short abbreviation should be replaced in the revised manuscript for better data presentation.

Response: The lengthy figure captions in Figure 2 were reduced and replaced with short abbreviation in the revised manuscript.

  1. The prepared composite materials should further be analyzed by Raman spectroscopy to confirm the successful conversion of starting materials and the different phases of individual components.

Response: The as-prepared composite materials have been analysed by Raman spectroscopy in the revised version, the results confirm the successful conversion of starting materials and the different phases of individual components. In Figure S1, the peaks located in 202, 223, 304, 321, and 350 cm−1, which corresponded to the characteristic peaks of Ni3S2 phase. Moreover, the CNS series sample have new peaks centered at 373, 450, 517, and 673 cm−1, indicating the existence of Co9S8.

  1. The performance of electrocatalysts depends on the active mass. The values should be mentioned in the experimental section of the revised manuscript.

Response: We added the values of active mass in the experimental section of the revised manuscript.

  1. The upper limits of X- & Y-axes of the EIS plot (Figure 6c) should be identical, for example, 8 (Ohm) in both axes. Further, the zoomed portion of EIS also should be included as an inset.

Response:  The upper limits of X- & Y-axes of the EIS plot (Figure 6c) were corrected to be identical, and unified to 8 (Ohm) in both axes. The zoomed portion of EIS was added as an inset.

  1. The figure quality should be improved in the revised manuscript, especially, Figure 1, 2, 6 and 7.

Response: Figures 1, 2, 6 and 7 were redrawn and the figure quality was improved in the revised manuscript.

  1. Plenty of cobalt & nickel sulfide-based electrocatalysts are reported in the literature. What are the advantages of the synthesized electrocatalysts, in terms of properties and electrocatalytic activity? These points should clearly be mentioned in the revised manuscript.

Response: The advantages of the synthesized electrocatalysts includes properties and electrocatalytic activity were emphasized in the revised version.

  1. The conclusion part of the manuscript is short. The important findings should be included in the revised manuscript.

Response: We added some important findings in the conclusion part of the revised manuscript.

Reviewer 3 Report

The results of the work are very promising and as the authors themselves say the approach in this work may provide the future rational design and synthesis of satisfactory OER electrocatalysts.

Author Response

Reviewer 3

Comments and Suggestions for Authors

The results of the work are very promising and as the authors themselves say the approach in this work may provide the future rational design and synthesis of satisfactory OER electrocatalysts.

Response: Thank you for your affirmation of our work.

Round 2

Reviewer 1 Report

The authors have now improved. I suggest accepting. Thank you

Reviewer 2 Report

The authors have revised the manuscript based on the reviewer's comments and the current version could be accepted for publication in the special issue.